# Ontology-based box embeddings and knowledge graphs for predicting phenotypic traits in *Saccharomyces cerevisiae*

**Filip Kronström**                                          FILIPKRO@CHALMERS.SE
**Daniel Brunnsåker**
*Chalmers University of Technology and University of Gothenburg, Sweden*

**Ievgeniia A. Tiukova**
*Chalmers University of Technology, Sweden*
*KTH Royal Institute of Technology, Sweden*

**Ross D. King**
*Chalmers University of Technology and University of Gothenburg, Sweden*
*University of Cambridge, United Kingdom*

**Editors:** Leilani H. Gilpin, Eleonora Giunchiglia, Pascal Hitzler, and Emile van Krieken

## Abstract

We present a method that uses graph neural networks (GNNs) to predict and interpret the effect of gene deletions in the yeast *Saccharomyces cerevisiae* from a knowledge graph (KG) with ontology-based box embeddings.

We construct the KG from community databases using terms defined in several ontologies. From the class hierarchies in the ontologies, box embeddings are learnt as low dimensional representations of the nodes in the graph, which are used together with GNNs to predict cell growth for double gene knockouts from the KG. With this we show that high level qualitative information can be used to predict experimental data.

Prediction performance was improved when using box embeddings of ontologies to represent the nodes in the graph, compared to learning features specific for this task. This suggests that class hierarchies in ontologies contain useful information about the domains, which can be extracted in the training of the box embeddings. We also demonstrate that our model can generalise beyond the task it was trained for by evaluating it on other types of genetic modifications.

Additionally, we apply model interpretability techniques to identify co-occurring edges critical for predictions. Our findings are further validated by a biological experiment that reveals an association between inositol utilisation and osmotic stress resistance, emphasising the model's potential to guide scientific discovery.

## 1. Introduction and Related work

A deeper understanding of cellular function and the roles of individual genes is central to biological research and critical for applications such as drug development. The yeast *Saccharomyces cerevisiae* (baker's yeast) is among the most extensively studied organisms. It has attracted research interest, not only due to its industrial applications, such as the production of beer, wine, and biofuels, but more importantly because it serves as a model eukaryotic organism. Through this role it helps our understanding of higher eukaryotes, such as humans and plants (Parapouli et al., 2020). Despite decades of research, our understanding of yeast biology is still incomplete: many genes remain unannotated (Wood et al., 2019), and interactions between genes can lead to complex and unexpected phenotypic outcomes (Costanzo et al., 2019). To improve our knowledge about any organism,

actual experiments in the lab play a crucial role. However, given the complexity of biological systems, the number of experiments needed to fully explore even the simplest of organisms is incredibly large. Because of this, methods supporting hypothesis generation at scale are highly useful to speed up new discoveries (King et al., 2004).

Decades of research on *S. cerevisiae* is available in literature, and also in more structured formats, such as databases. Saccharomyces Genome Database (SGD) aggregates curated information about *S. cerevisiae* genes. It contains Gene Ontology-annotations and observed phenotypes for genes, as well as regulations and interactions between genetic strains (Engel et al., 2024). Information about reactions (biochemical events where a substrate is converted into a product) and pathways (a series of interconnected reactions that collectively drive cellular functions) is available in, among others, BioCyc (Karp et al., 2019).

Information in such databases is often represented and communicated using ontologies and controlled vocabularies. The Gene Ontology (GO) defines classes describing processes, functions, and components in cells (Ashburner et al., 2000). To properly represent phenotypes in SGD the Ascomycete Phenotype Ontology (APO) was developed (Costanzo et al., 2009). Phenotypes describe observed characteristics resulting from the interaction between a genotype and an environment, such as growth characteristics or resistances to environmental or chemical perturbants. Chemical compounds are specified in Chemical Entities of Biological Interest (ChEBI) (Hastings et al., 2016). The Interaction Network Ontology (INO) (Hur et al., 2015) and Molecular Interactions (MI) (Hermjakob et al., 2004) defines genetic, physical and regulatory interactions between genes and proteins. Commonly used relations between classes are introduced in the Relations Ontology (RO) (Mungall et al., 2020). The Basic Formal Ontology (BFO) is a top-level ontology developed to simplify alignment of terms from different ontologies (Arp et al., 2015).

Knowledge graphs (KGs) have successfully been used to describe heterogeneous data from various domains by combining instantiated facts with semantically meaningful concepts from ontologies. In the biomedical domain KGs like BioKG (Walsh et al., 2020) and SPOKE (Morris et al., 2023) combine information from different databases to create one large heterogeneous graph. There are also graphs describing narrower phenomenon such as the protein-protein associations and the drug-drug interactions in the Open Graph Benchmark (Hu et al., 2020).

Embeddings of ontologies or knowledge graphs in an $n$-dimensional space, $\mathbb{R}^n$, where its structure is in some way maintained, have proved useful for downstream tasks such as link and property prediction from KGs. TransE represents links between entities as translations in a vector space (Bordes et al., 2013). TransE has later been extended to make use of hierarchical class information in $\mathcal{EL}^{++}$ ontologies to represent 'subClassOf' relationships as classes maintained within hyperspheres (Kulmanov et al., 2019) and axis-aligned hyperrectangles (Peng et al., 2022). Gumbel boxes have been introduced, where min and max Gumbel distributions are used to represent box parameters, to avoid large flat regions of the loss landscape for transitive relation embeddings (Dasgupta et al., 2020). Instead of representing relations as translations of classes, graph neural networks (GNNs), e.g. GraphSAGE, can be used to aggregate features from neighbors in the graph to generate embeddings of nodes (Hamilton et al., 2017).

Predicting biological properties from structured background knowledge can be done in different ways. Ma et al. (2018) encode GO-annotations together with the GO hierarchy in a

neural network to predict cellular growth in *S. cerevisiae*. By predicting protein abundances using mined patterns from a Datalog knowledgebase containing facts from databases such as SGD, Brunnsåker et al. (2024) connected qualitative concepts to quantified intracellular measurements. KG embeddings have, for example, been used by Gualdi et al. (2024) to predict genes associated with diseases from a protein interaction KG.

## 2. Material and methods

### 2.1. Knowledge graph

We have created a heterogeneous knowledge graph describing genes in the yeast *Saccharomyces cerevisiae*, by combining facts expressed in classes and relations from multiple ontologies. The graph is specified in description logic, only using TBox statements by rewriting class assertions, $C(a)$, as $\{a\} \sqsubseteq C$ and role assertions, $r(a, b)$, as $\{a\} \sqsubseteq \exists r.\{b\}$. In this way, we get the same representation of asserted facts from databases as we have for terminological statements from ontologies, like GO or ChEBI. This simplifies the interface between KG, box embeddings, and GNN, introduced in Sections 2.2 and 2.3.

The knowledge graph is created from data in SGD, where the information is defined using terms from several different ontologies, either using the structure from BFO or can easily be merged with it. A high level overview of the graph, showing how different node types are connected, can be seen in Figure 1a. In Figure 1b we show examples of the hierarchies classes instantiating these nodes are represented in.

The GO-annotations in SGD are naturally described by classes in the Gene Ontology and relations from the OBO Relations Ontology, which are specified in the database. Phenotypes are described using terms from APO where a phenotype is represented by an 'observable', for example 'heat sensitivity', and possibly a 'qualifier', for example 'increased'. We represent the phenotype as the subclass of the intersection of these two types of classes, and phenotypes are linked to genes using the RO relation 'has phenotype'. Some phenotypes describe observables related to specific chemicals, in such cases the chemical class in ChEBI is linked with a custom relation, 'aboutChemical'. To form a closer connection between genes and chemicals related to phenotypes, which proved useful for downstream tasks (see Section 2.3), a link specific to the type of observable was added between the gene and the chemical. An example of how this is implemented in description logic can be seen in (2) in Appendix A.

Gene regulation in SGD is a directed relationship between two genes that can be positive, negative, or unspecified, and of different types, for example, regulation of protein activity or expression. In some instances, a biological process from GO specifies under which conditions the regulation occurs. We introduce custom relations describing regulation type and direction, which we use to link the two genes in the graph. When a biological process is specified we also link the genes to a gene-specific subclass of the 'regulation' class from INO, which in turn is linked to the GO-term. The description logic implementation of such a regulation can be seen in (3) in Appendix A.

Interactions between genes are undirected relationships, sometimes connected to a phenotype, observed together with the interaction. Similarly to regulation this is modelled as a link between the involved genes and a gene specific subclass of either a protein-protein

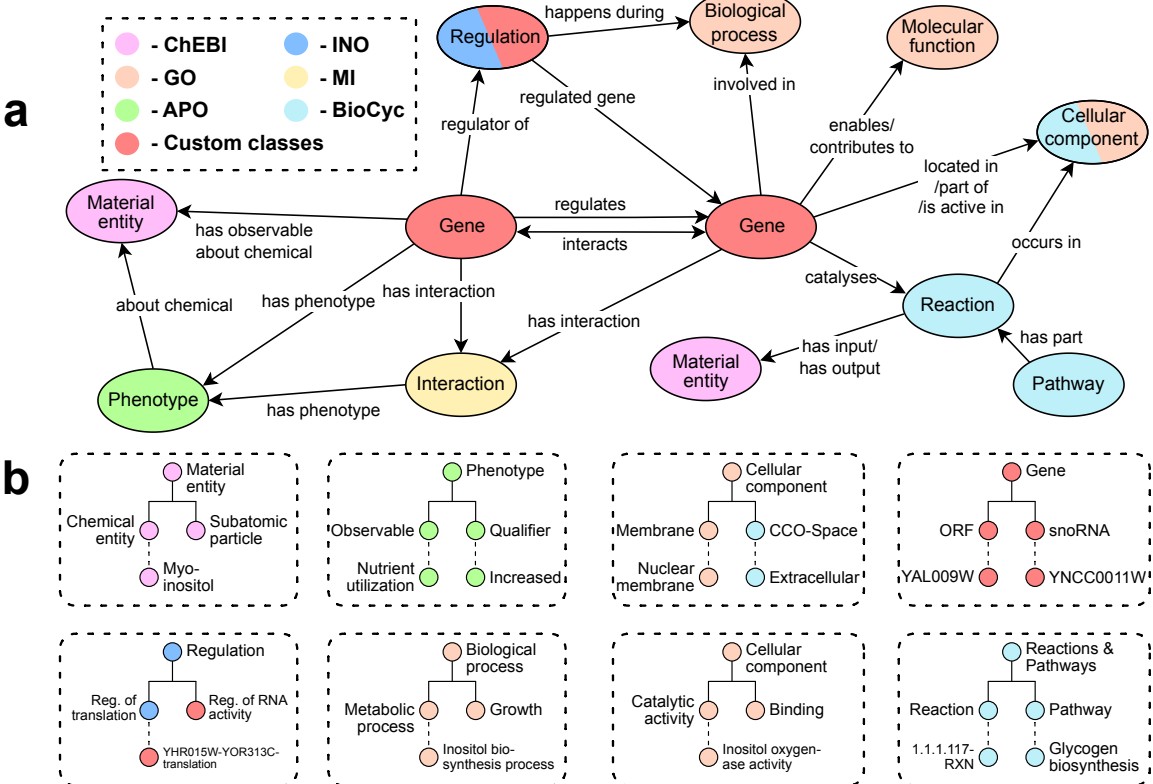

Figure 1: An overview of the different types of classes and how they are connected in the knowledge graph is shown in **a**. The color of the nodes specifies where the classes are defined. **b** shows examples from the hierarchies defining classes in the domains introduced in Section 2.2.

`interaction` from INO or a `genetic interaction` from MI, which is linked to the phenotype.

Beyond the data from SGD we have also included information about reactions and pathways from BioCyc, which uses its own controlled vocabulary. In the graph, reactions are linked to their input and output chemicals, as well as, when specified, genes they are catalysed by and locations in the cell where they take place. We link pathways to their involved reactions, as well as to the compounds that are consumed and produced.

### 2.2. Class embeddings

As the classes in our KG have a well defined hierarchy from the ontologies, we use axis-aligned hyperrectangles, box embeddings, to represent them. We interpret the '`subClassOf`' relation as a class-box being contained within its superclass-box. Note that the box embeddings only capture the class hierarchies and no other roles present in the ontologies or KG. Relations, other than `subClassOf`, are instead modelled by the GNN described in Section 2.3.

As proposed by Dasgupta et al. (2020) we use Gumbel boxes which are learnt by minimising the binary cross-entropy (BCE),

$$\mathcal{L}_{\text{Box}}(c, p) = BCE\left(\texttt{subClassOf}(c, p), \frac{\textsf{Vol}(\text{Box}(c) \cap \text{Box}(p))}{\textsf{Vol}(\text{Box}(c))}\right) + \lambda \left\|\text{Boxes}\right\|^2, \qquad (1)$$

where $\texttt{subClassOf}(c, p)$ is 1 if $c$ is a subclass of $p$ and 0 otherwise (negative example). $\textsf{Vol}$ is the softplus approximation of the volume of a box, $\cap$ is the expected intersection of Gumbel boxes, and $\lambda$ controls the regularisation of the box sizes. To implement this we use the `box-embeddings` package (Chheda et al., 2021). Negative examples are generated by drawing random classes, $\bar{p}$, that are not in the set of parents to $c$, i.e. not in $\{p|c \sqsubseteq^* p\}$[1].

We divide our classes into eight different domains, seen in Figure 1b, for which separate box embeddings are found. These splits generally align well with the ontologies the classes are from, or disjoint branches in the same ontology. The reasoning behind this is that these domains represent non-overlapping concepts, so not much is gained by representing them in the same embedding space. Doing this also allows us to reduce the dimensionality of the embedding space and vary it depending on the number of classes in the domain.

Parameters used to train the box embeddings and the dimensions of the different domains are reported in Appendix B.1.

### 2.3. Prediction models

To demonstrate the usefulness of our KG we train a GNN to predict phenotypic traits in *S. cerevisiae* from it. We use data from Costanzo et al. (2016) where cell growth is measured when pairs of genes are deleted (digenic deletions) from the genome. By comparing this to the cell growth when no genes are removed a fitness score can be found, describing how the deletion of the two genes affect growth. A subset of this data, grown under the same standard experimental conditions (30°C), is used to train our model. This results in a data set with 10,085,183 examples of deleted gene pairs and a corresponding fitness. Note that the genetic interaction relation from SGD describes a similar phenomena, often derived from the same dataset. These relations are thus removed from the graph before training to avoid data leakage.

We use the box embeddings, introduced in Section 2.2, describing the class hierarchies, as node features. This results in a heterogeneous graph where nodes belong to the eight domains seen in Figure 1b, with features co-existing within the domains. Adding reverse links to the graph to allow for message passing in both directions results in 204 different types of links. After removing infrequent (<1,000) and overlapping edges we end up with 72 different types of links used for prediction.

An overview of the pipeline predicting gene-pair fitness is shown in Figure 2a. We employ a mean-aggregated, heterogeneous variant of the GraphSAGE embedding algorithm introduced by Hamilton et al. (2017) to learn node representations within the KG, which has previously shown promise in KG- and network-related prediction tasks (Ma et al., 2023; Syama et al., 2023; Vretinaris et al., 2021). The dimensionality of the SAGE message-passing modules is adjusted based on the type of source-edge-target triple, and specifically varies with the number of edges directed toward the target domain. Domains with a high degree of incoming connectivity, such as 'Material entities' or 'Genes', are assigned higher

---

1. This means negative examples will push towards disjoint boxes which is, due to the open world assumption, not necessarily the true representation, but a way to produce a box embedding discriminating between classes.

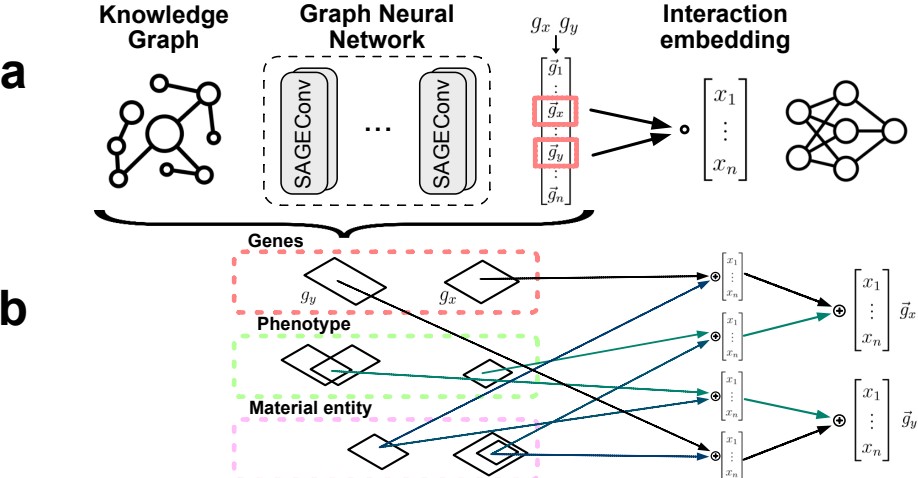

Figure 2: An overview of the system predicting the fitness when deleting pairs of genes is shown in **a**. **b** shows how classes in the different domains are represented by boxes and how information is aggregated in the GNN. Arrows from the boxes represent learnable SAGE modules, different for each `source domain-edge-target domain` type. The fitness is predicted from the Hadamard product of the embeddings of the two deleted genes.

dimensional feature spaces. The resulting class embeddings, generated by the GNN, capture aggregated neighbourhood information but are no longer in the form of box embeddings. Figure 2b illustrates how information is propagated from the initial box embeddings across different domains to the gene embeddings. To predict the fitness of a gene pair, we compute the Hadamard product of their embedding vectors and feed the result into a fully connected neural network, which outputs a real-valued prediction. The model is trained end-to-end by minimising the mean squared error using the Adam optimiser.

The models are trained and evaluated using 10-fold cross validation where the data split is based on the genes, discarding pairs made up of both training and validation genes. As a result, no pairs involving any of the genes in the validation set are seen during training, meaning representations found for individual genes in the training data does not affect the evaluated predictions.

Hyperparameters, such as learning rate, regularisation $\lambda$, depth and width of the fully connected neural network, depth of GNN, and embedding dimensions throughout the GNN (these are then doubled for the most common target domains, and halved for the least common), are tuned using Bayesian optimisation on a different data split than the one evaluated in Section 3.1. The used hyperparameters are reported in Appendix B.2.

## 3. Results

### 3.1. Gene deletion fitness prediction

In Table 1 we present the coefficient of determination ($R^2$) for the model described in Section 2.3, with and without the box embeddings introduced in Section 2.2. When not using box embeddings we instead learn shallow embeddings to represent the classes in

the KG. We also compare to a Light Gradient Boosting Machine (LightGBM) (Ke et al., 2017) on the instantiation of the phenotype information from the KG. The phenotypes describe observable characteristics of the genes and is the part of the KG we expect to be most informative for this task (further support for this is seen when considering feature importances for the GNN, mentioned in Section 3.2, which are dominated by phenotypes). The instantiation of the phenotypes is sparse with 2680 features.

Table 1: Results from 10-fold cross-validation of digenic deletion fitness. Both GNN models share the same architecture but differ in class representations: one uses box embeddings for ontology hierarchies, while the other employs task-specific shallow embeddings. The instantiation model uses a sparse feature matrix with non-zero entries for phenotype annotations from the KG. All pairwise model differences are significant ($p<0.05$, paired t-test).

| Model | Mean $R^2$ | SD |
|---|---|---|
| GNN with box embeddings | **0.360** | 0.043 |
| GNN without box embeddings | 0.329 | 0.043 |
| Instantiations + LightGBM | 0.211 | 0.022 |

From the results it is clear that the GNN generates a gene embedding which can be used for predicting this fitness to a reasonable degree, given the amount of noise typically present in biological measurements (Li et al., 2021). Using the box embeddings to represent classes rather than learning these from scratch results in a significant ($p<0.05$, paired t-test) improvement, and the instantiated phenotype information seems to be useful for prediction, but is not as informative as the full KG.

The parity plot for the predictions can be seen in Figure 3(a). From this we can see that most double gene deletions do not have major impact on the fitness. We can also see a clear shrinkage effect where the model mispredicts extreme values, especially deletions with low fitness are overestimated.

Kuzmin et al. (2018) performed a similar study to the one we used for training our models, but with trigenic deletion fitness. We use this data, a total of 15,095 triple deletion datapoints, to evaluate our model on this modified version of the original task. For this we use one model trained on the full dataset from Costanzo et al. (2016), but instead perform the Hadamard product between the three involved genes. Notably we here achieve an $R^2$ of 0.380, slightly higher than on average achieved in the cross validation of digenic deletions. The parity plot, seen in Figure 3(b), shows promise in generalising to a new task. An important note on this experiment is that, unlike the double deletion experiment, the individual genes making up the triple deletions are now seen as parts of double deletion examples in training.

## 3.2. Model interpretation and experimental evaluation

Since our predictions stem from a KG where each link holds domain-relevant meaning, we explore patterns among important edges. We use an input-gradient method (Shrikumar et al., 2017) via Captum (Kokhlikyan et al., 2020) to attribute edge importance. By multiplying the individual importance scores for the involved genes we get a measure of the importance of co-occurring edges. Summing these values for all predictions gives a global

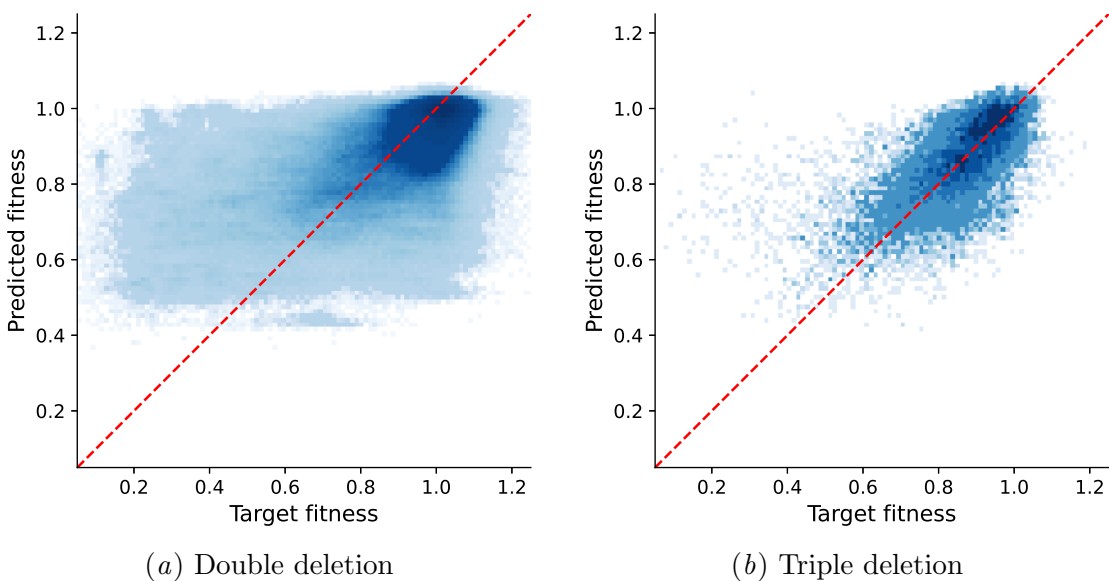

(*a*) Double deletion

(*b*) Triple deletion

Figure 3: Parity plots for double, (*a*), and triple, (*b*), gene deletions. For the double deletion, the predictions from all validation sets in the cross validation are shown.

importance of such edge-pairs. This can be interpreted as the impact of the interaction between the two traits on the fitness.

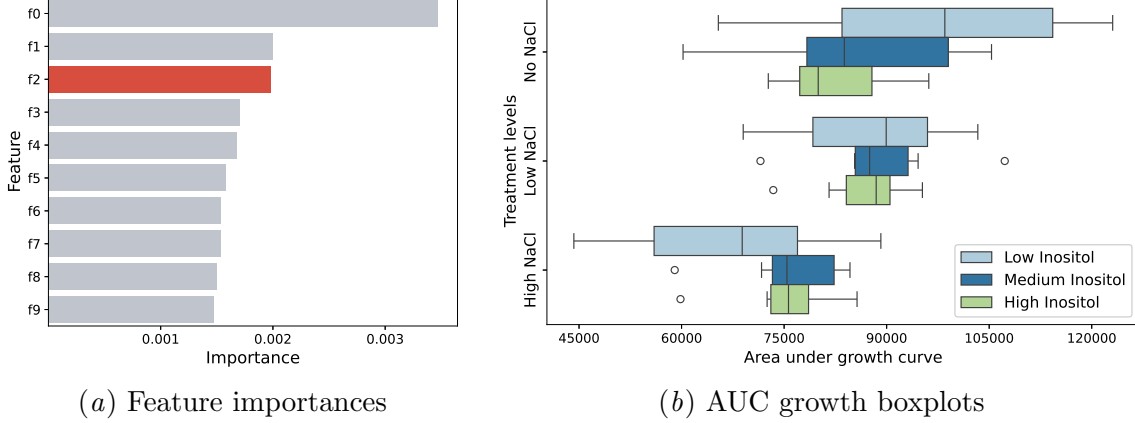

(*a*) Feature importances

(*b*) AUC growth boxplots

Figure 4: An overview of the selection and results of the experiment we performed. (*a*) shows the highest ranked importances of edge-pairs and the pair selected for the experiment, nutrient utilisation of inositol and stress resistance to NaCl, is highlighted in red. *f0* and *f1*, which have a higher assigned weight, are discarded due to safety and lab constraints as it involves the chemical bleomycin. (*b*) Box plot showing the distribution of AUC for all of the experimental conditions tested. Inositol supplementation significantly impacts growth dynamics in high doses ($p < 0.05$). NaCl stress changes the impact of inositol in a dose dependent manner, suggesting an interactive effect ($p < 0.05$).

To identify patterns corresponding to viable experiments for standard lab setups we filtered for edges related to nutrient utilisation phenotypes. A more detailed description of the filtering process can be found in Appendix C.1. The top 10 most important edge pairs are shown in Figure 4(a) and detailed in Appendix C.2. The highest-weighted, safely testable pair was selected and highlighted in red in Figure 4(a), linking one of the involved genes to inositol (vitamin B8) utilisation and the other to NaCl stress resistance, suggesting a potential interaction between these traits.

To experimentally test this hypothesis, a perturbation experiment was performed in an automated laboratory cell (Williams et al., 2015), in which inositol and NaCl was supplied in a range of concentrations, details about the experimental design and cultivation methods can be found in Appendix C.3. An $\Delta ino1$ mutant (INOsitol requiring) was used for all subsequent experiments, as it is unable to synthesise inositol on its own, ensuring that intracellular was acquired only through transport from the media. The growth dynamics of the cells in the different experimental conditions were summarised with the area under curve (AUC) of the growth curves, providing a single-valued measure of the biomass accumulation over the course of the experiment. The full growth dynamics can be seen in Figure 5 in Appendix C.4 and summarising boxplots are shown in Figure 4(b). Statistical testing for interaction effects was done with a Gaussian generalised linear model (GLM), further details can be found in Appendix C.4.

Table 2: Estimated parameters from the GLM examining the effects of myo-Inositol-supplementation and NaCl treatment on growth dynamics. The table presents coefficient estimates, $p$-values and confidence intervals for the main effects and interaction terms. Significant interactions indicate that the effect of myo-inositol supplementation changes depending on treatment levels. The two highlighted rows indicate the significant interaction effect.

| | Coefficient ($\times 10^3$) | Confidence interval ($\times 10^3$) | $p$-value |
|---|---|---|---|
| Intercept | 97.29 | [88.20, 106] | 0.000 |
| Medium inositol | -11.84 | [-24.7, 1.02] | 0.071 |
| High inositol | -14.56 | [-27.9, -1.24] | **0.032** |
| Low NaCl | -9.50 | [-22.4, 3.37] | 0.148 |
| High NaCl | -30.61 | [-43.5, -17.8] | **0.000** |
| Medium inositol × Low NaCl | 13.11 | [-5.40, 31.6] | 0.165 |
| High inositol × Low NaCl | 13.38 | [-5.45, 32.2] | 0.164 |
| Medium inositol × High NaCl | 20.92 | [2.41, 39.4] | **0.027** |
| High inositol × High NaCl | 22.64 | [3.40, 41.9] | **0.021** |

These empirical results, seen in Figure 4(b) and Table 2, indicate a significant interaction between inositol supplementation and induced NaCl stress, verifying that the proposed edge-interactions are consistent with experimental data. Specifically, supplementing with inositol rescued cells from NaCl-induced stress, indicating that inositol availability enhances their ability to withstand salt stress. Inositol has previously been implicated in biosynthesis and integrity of cell membranes (Culbertson and Henry, 1975). Since NaCl can disrupt osmotic balance, enhanced membrane stability is likely to have a protective effect for the cells.

## 4. Discussion

We constructed a knowledge graph (KG) describing *S. cerevisiae* genes and demonstrated its biological relevance by predicting digenic deletion fitness. While the predictive $R^2$ of 0.36 may seem low, biological data is inherently noisy, and even replicating experiments is challenging (Roper et al., 2022). Moreover, our model predicts quantitative outcomes from high-level qualitative information.

This capability enabled interpretability techniques to guide experiment selection, underscoring the value of structured data representation and computational methods in accelerating research. Our edge filtering for viable experiments introduces biases regarding the type of hypotheses generated. Leveraging large language models could be one approach to automatically refine this selection and reveal overlooked experiments.

The proposed model generates an embedding of the nodes in the KG, using the fitness data as a biological signal to guide the training. We found they can be used for a slightly different task than they were trained for, by evaluating trigenic gene deletions. This is likely, at least partly, due to the individual genes involved no longer being unseen during training. The fitness will depend heavily on the traits of the individual genes, which will be better represented for genes in the training data. The embeddings could potentially be applied to a broader range of tasks, such as GO annotation of genes, where multiple knowledge sources typically are integrated (Merino et al., 2022).

We have not utilised sequence information, despite it being the most informative data about genes and fully available for *S. cerevisiae*. Our current setup relies on box embeddings from a rudimentary SGD gene categorisation, which offers little discrimination, with over 90% of genes falling into the same category. Incorporating sequence-based representations would provide richer and more meaningful gene embeddings.

Class hierarchies play a central role in our data representation, enabling unseen terms to contribute if well-placed in the hierarchy. A key advantage of this approach is its interpretability for domain experts compared to more complex symbolic representations. Our experiments suggest that leveraging hierarchies improves performance. However, we currently use them only as a prior class representation without explicitly maintaining them in the GNN. Future work could ensure box representations propagate hierarchy constraints, potentially providing guarantees for the resulting embeddings.

## 5. Conclusion

With this work we show how publicly available data can be used to construct a KG describing *S. cerevisiae*. We show that we can train models to predict biological measurements from such a KG, and that they can generalise beyond the task they were trained for. Further, we use interpretability tools to form a hypothesis about phenotype interactions in *S. cerevisiae* which we find support for by performing a biological experiment, uncovering an association between inositol utilisation and NaCl stress. This illustrates how models with semantic grounding can help in scientific discovery.

The code and data for this project is available at `https://github.com/filipkro/kg-box-emb`

## Acknowledgments

The computations were enabled by resources provided by the National Academic Infrastructure for Supercomputing in Sweden (NAISS), partially funded by the Swedish Research Council through grant agreement no. 2022-06725. This work was supported by the Wallenberg AI, Autonomous Systems and Software Program (WASP) funded by the Alice Wallenberg Foundation, the UK Engineering and Physical Sciences Research Council (EPSRC) [EP/R022925/2, EP/W004801/1 and EP/X032418/1], the Chalmers AI Research Centre, and Swedish Research Council Formas [2020-01690].

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

## Appendix A. Examples of description logic in KG

A description logic example of how a phenotype with a qualifier and chemical are specified in the KG. This example is about decreased (`APO_0000003`) utilisation of carbon source (`APO_0000096`) of lactate (`CHEBI_16004`), observed for the gene `YBL030C`.

$$
\begin{aligned}
&\texttt{APO\_0000098-APO\_0000003-CHEBI\_16004} \sqsubseteq \texttt{APO\_0000098} \sqcap \texttt{APO\_0000003} \\
&\texttt{APO\_0000098-APO\_0000003-CHEBI\_16004} \sqsubseteq \exists \texttt{aboutChemical.CHEBI\_16004} \\
&\texttt{YBL030C} \sqsubseteq \exists \texttt{RO\_0002200.APO\_0000098-APO\_0000003-CHEBI\_16004} \\
&\texttt{YBL030C} \sqsubseteq \exists \texttt{hasChemNutrientUtilization\_Decreased.CHEBI\_16004}.
\end{aligned} \tag{2}
$$

A description logic example of how a gene (`YCR073C`) is positively regulating the protein activity (`INO_0000104`) of another gene (`YLR113W`). This regulation happens during (`RO_0002092`) cellular response to heat (`GO_0034605`).

$$
\begin{aligned}
&\texttt{YCR073C-YLR113W-protein\_activity-positive} \sqsubseteq \texttt{INO\_0000104} \\
&\texttt{YCR073C} \sqsubseteq \exists \texttt{positive\_regulator\_of.YCR073C-YLR113W-protein\_activity-positive} \\
&\texttt{positive\_regulator\_of.YCR073C-YLR113W-protein\_activity-positive} \\
&\qquad \sqsubseteq \exists \texttt{regulated\_gene.YLR113W} \\
&\texttt{positive\_regulator\_of.YCR073C-YLR113W-protein\_activity-positive} \\
&\qquad \sqsubseteq \exists \texttt{RO\_0002092.GO\_0034605} \\
&\texttt{YCR073C} \sqsubseteq \exists \texttt{positively\_regulating.YLR113W}.
\end{aligned} \tag{3}
$$

## Appendix B. Hyperparameters

### B.1. Box embedding parameters

Table 3: Parameters used for the box embeddings of the different domains

| Domain | Dimensions | Epochs | Lr | Regularisation | Gumbel temperature | Neg. ex. ratio |
|---|---|---|---|---|---|---|
| Material entity | 10 | 1,000 | 1e-2 | 1e-3 | 0.25 | 2.0 |
| Genes | 8 | 600 | 1e-2 | 1e-3 | 0.25 | 4.0 |
| Regulations | 5 | 500 | 1e-2 | 1e-3 | 0.25 | 2.0 |
| Molecular functions | 5 | 500 | 1e-2 | 1e-3 | 0.25 | 2.0 |
| Biological processes | 5 | 500 | 1e-2 | 1e-3 | 0.25 | 2.0 |
| Phenotypes | 4 | 500 | 1e-2 | 1e-3 | 0.25 | 2.0 |
| Reactions & Pathways | 4 | 500 | 1e-2 | 1e-3 | 0.25 | 2.0 |
| Cellular components | 4 | 500 | 1e-2 | 1e-3 | 0.25 | 2.0 |

### B.2. Prediction model hyperparameters

The best performing model was trained for 500 epochs, with a learning rate of 1e-5, and L2 regularisation weight of 0.1. The depth of the GNN was 2 and the embedding dimensions for the domains are listed in Table 4 and are the same throughout the GNN. The fully connected neural network predicting the interaction from the embeddings is of depth 3 with 64, 8, and 1 neurons respectively.

Table 4: Embedding dimensions for the different domains throughout the GNN.

| Embedding dimensions | 32 | 64 | 128 |
|---|---|---|---|
| Domains | Cellular components Molecular functions Reactions Regulations | Biological processes Phenotypes | Material entities Genes |

## Appendix C. Model-driven experiment

### C.1. Edge filtering

Table 5: We filter for co-occurring edge pairs in which at least one edge connects a gene to a node that is a subclass of one of the following APO classes, related to nutrient utilisation.

| APO Class | Description |
|---|---|
| APO_0000096 | General nutrient utilisation |
| APO_0000097 | Auxotrophy |
| APO_0000099 | Utilisation of nitrogen source |
| APO_0000100 | Nutrient uptake |
| APO_0000125 | Utilisation of phosphorous source |
| APO_0000219 | Utilisation of sulfur source |

Table 6: We also allow edge pairs where at least one of the edges links a gene to a chemical through any of the following relations.

```
hasChemNutrientUtilization
hasChemNutrientUtilization_Increased
hasChemNutrientUtilization_Decreased
```

## C.2. Top edge pairs

Table 7: The 10 edge pairs with the highest importance weight after filtering for the criteria specified in Appendix C.1. The edge pair selected for the experiment is highlighted. `Ch.Nutr.Util.` is short for `hasChemNutrientUtilization`, `Ch.Nutr.Util.Dec.` is short for `hasChemNutrientUtilization_Decreased`, and `Ch.StressRes.` is short for `hasChemStressResistance`.

| Importance | Relation1 | Class1 | Relation2 | Class2 |
|---|---|---|---|---|
| 0.003471 | Ch.Nutr.Util. | CHEBI_17268 | Ch.StressRes. | CHEBI_22907 |
| 0.002002 | Ch.StressRes. | CHEBI_22907 | has_phenotype | APO_0000099-APO_0000245-CHEBI_14321 |
| **0.001985** | **Ch.Nutr.Util.** | **CHEBI_17268** | **Ch.StressRes.** | **CHEBI_26710** |
| 0.001705 | Ch.Nutr.Util.Dec. | CHEBI_23414 | Ch.StressRes. | CHEBI_22907 |
| 0.001679 | hasChemCellMorph | CHEBI_26710 | has_phenotype | APO_0000099-APO_0000245-CHEBI_14321 |
| 0.001580 | Ch.Nutr.Util. | CHEBI_17268 | Ch.StressRes. | CHEBI_50145 |
| 0.001541 | Ch.Nutr.Util.Dec. | CHEBI_77995 | Ch.StressRes. | CHEBI_49470 |
| 0.001537 | Ch.StressRes. | CHEBI_22907 | has_phenotype | APO_0000099-APO_0000245-CHEBI_26271 |
| 0.001500 | Ch.Nutr.Util. | CHEBI_17268 | has_phenotype | APO_0000059-APO_0000002-CHEBI_26710 |
| 0.001477 | Ch.Nutr.Util.Dec. | CHEBI_16236 | Ch.StressRes. | CHEBI_22907 |

## C.3. Cultivation method

The $\Delta ino1$ deletion mutant was taken from the EUROSCARF deletion collection, with the strain background being BY4741, genotype: MATa, $his3\Delta1$, $leu2\Delta0$, $met15\Delta0$, $ura3\Delta0$ (Y01272).

The $\Delta ino1$ mutant was pre-cultured overnight in minimally buffered delft media containing the following: 5g/L (NH4)2SO4, 3g/L KH2PO4, 0.5g/L MGSO4. 7H2O, and 1mL/L trace metal and vitamin solutions as described by Verduyn et al. (1992), 25 mg/L myo-inositol and 2% glucose (w/v) in 30°C, and 220rpm. The pre-culture was adjusted to 0.5 OD600, and robotically dispensed with a 1:20 dilution into a 96-well microculture plate using a Hamilton Microlab Star liquid handling robot. A negative control was also included to assess the baseline growth of the $\Delta ino1$ mutant without any supplementation of myo-inositol. Additionally, myo-inositol-free media with 0.25% (w/v) glucose, myo-inositol (Sigma aldrich 57570-100G), Sodium chloride (Merck 1064041000) and MilliQ-water was robotically dispensed, resulting in a total volume of $250\mu L$ and the concentrations defined in Table 8.

Table 8: The concentrations of inositol and NaCl used for the experiment.

| Inositol | NaCl |
|---|---|
| 0.00 $m$Molar | 0.0 Molar |
| 0.01 $m$Molar | 0.0 Molar |
| 0.01 $m$Molar | 0.3 Molar |
| 0.01 $m$Molar | 0.6 Molar |
| 0.05 $m$Molar | 0.0 Molar |
| 0.05 $m$Molar | 0.3 Molar |
| 0.05 $m$Molar | 0.6 Molar |
| 0.25 $m$Molar | 0.0 Molar |
| 0.25 $m$Molar | 0.3 Molar |
| 0.25 $m$Molar | 0.6 Molar |

A robust plate layout was generated with PLAID (Francisco Rodríguez et al., 2023). The processed plate was cultivated in the automated laboratory cell Eve. The plate was transferred from an automated incubator (30°C) to a Teleshaker Magnetic Shaking System, where it was shaken for 30s at 800 rpm, divided evenly between clockwise and counter-clockwise double-orbital shaking. After shaking, the plate was transferred to a BMG Polarstar plate reader, where it underwent optical density measurements at 600 nM (the temperature in the plate reader was kept at a constant 30°C). After measuring, the plate was returned to the incubator. The protocol was automatically repeated every 20 min for up to 24 h.

### C.4. Growth data processing and statistical testing

Outliers in the growth curves (measured through optical density at 600nm) were identified and filtered using the interquartile range (IQR), where any data points outside the range of [Q1-1.5 IQR, Q3+1.5 IQR] were excluded from the dataset. The filtered curves were then subsequently smoothed using a rolling mean of window size 3. The resulting averaged growth curves can be seen in Figure 5. Area under curve was calculated using `numpy.trapz` (`v1.26.4`). To assess the effects of inositol and NaCl on AUC, a generalised linear model was employed (`statsmodels v0.14.4`). The model was fitted using a Gaussian family distribution. Choice $\alpha$-value was set at 0.05. We modelled all factors as categorical to avoid imposing any assumptions on linearity. The model is specified as follows:

$$\text{AUC} \backsim C(Inositol) \times C(NaCl). \tag{4}$$

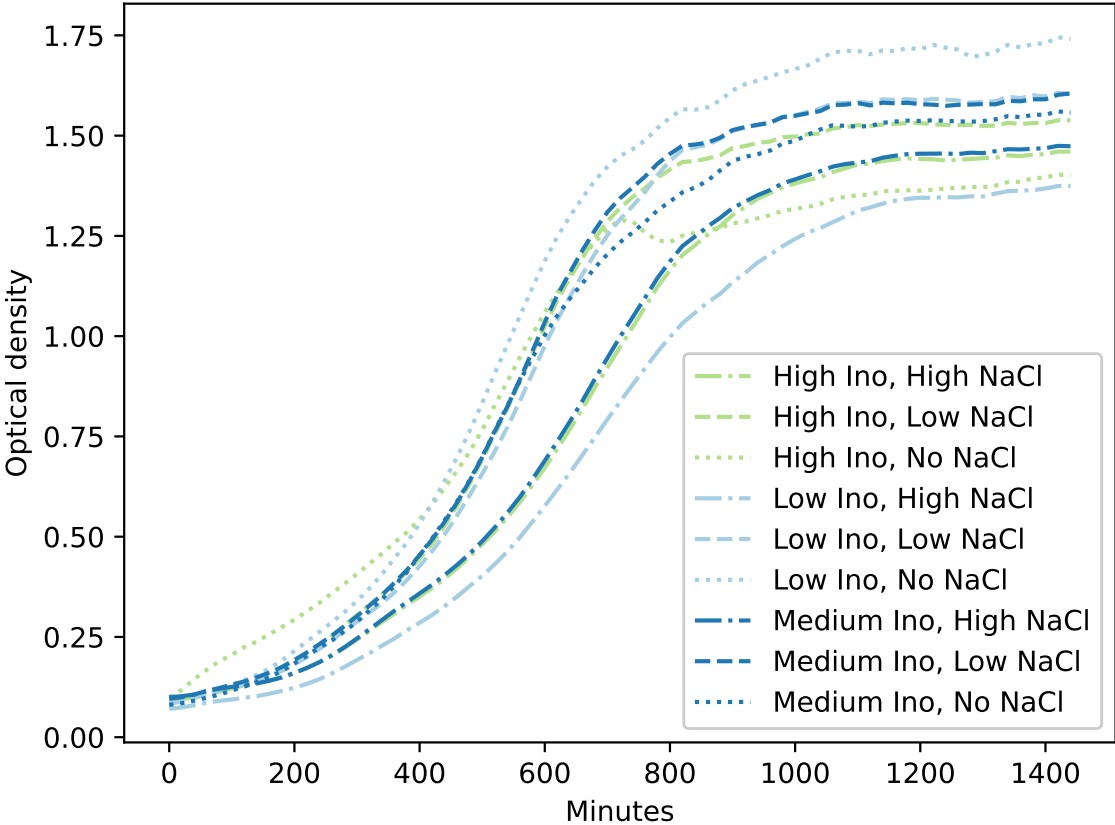

Figure 5: Growth curves showing the mean optical densities of the 6-8 repetitions for the different experimental groups. Optical density (at 600nM) is a unitless measurement typically used as an indirect measure of cell density and biomass.

