# OpenReview forum: "Ontology-based box embeddings and knowledge graphs for predicting phenotypic traits in Saccharomyces cerevisiae"
_nesyconf.org/NeSy/2025/Conference — NeSy 2025 Poster_

### Official Review · Reviewer_CNni · 2025-04-07
**Ontology-based box embeddings and knowledge graphs for predicting phenotypic traits in Saccharomyces cerevisiae**

**Rating:** 8
**Confidence:** 4

**Review:**

This article presents an implementation of a knowledge graph together with ontology-based box embedding for predicting attributes of mutations, in particular their impact on fitness. Overall the paper is a fascinating case study in building a knowledge graph and then using it to drive predictions.

Nevertheless, the article does not fully justify the choice of architecture and leaves some questions open which it would benefit answering in the paper. Why graph neural networks - what is the added value of other architectures? Why box embeddings - how are the boxes subsequently used, and what is their added value? Which parts of the heterogeneous knowledge graph contributed the most to the predictive performance?

Furthermore, I wonder how the dual and triple gene fitness outcome prediction is handled for the data splitting into test/train/validate. Is the data splitting stratified by combinations of genes that then are predicted together? If this is not the case, what is the risk of information leakage if some information for a gene is present in the training data?

Minor:
- Note capitalisation of "InChI" (in the paper one capitalised 'I" is missed)

**Anonymity:**

Remain anonymous

---

### Official Review · Reviewer_jfVG · 2025-04-08
**Interesting interdisciplinary paper, but clarity should be improved**

**Rating:** 7
**Confidence:** 4

**Review:**

The paper uses a Gumbel box embedding to represent classes of several biochemical ontologies like ChEBI and the Gene ontology. Different ontologies are embedded differently, also with different dimensions. Based on the ontology, from the data in the Saccharomyces Genome Database, a knowledge graph is built. The knowledge graph is then used to train a graph neural network that predicts gene interaction phenotypes in Saccharomyces cerevisiae (yeast), and in particular digenic deletion fitness. This is done using the Hadamard product of the gene embeddings, followed by a fully-connected neural network. The highest-weighted pair of genes was selected and the interaction tested using an automated lab. The lab experiments could plausibly confirm the predicted interaction.


The paper is interdisciplinary and combines several bio-chemical ontologies, a knowledge graph, box embeddings (via deep learning), graph neural networks and biochemical lab experiments. This is remarkable. The approach seems to be novel and is significant for NeSy, because a sucessful interaction between symbolic knowledge representation and deep neural networks is shown.
The paper is mostly well readable, but at some points, I would have preferred more clarity:

1. An overall architecture diagram would help to keep the overview.

2. It is not clear to me why the knowlegde graph is specified as an OWL TBox. I would expect an RDF graph, or at least an OWL ABox, especially because only class and role assertions are used.
From Figure 1 I get the impression that the knowledge graph conceptually is more an ontology, because does not contain specific individuals and their relations (as a KG usually does), but classes and their relations (as an ontology usually does). On the other hand, since data from the Saccharomyces Genome Database is used, I guess that in the end, it will actually be a knowledge graph.

3. It is not clear to me how the different ontologies are aligned when using a different box embedding per ontology. For example, how is phosphate (CHEBI:43474) related to "phosphate ion transport" (GO:0006817)? In a global box embedding, one would expect some geometric relation induced by the semantic relation of these terms. The paper says on p.4 "To align chemical compounds, ChEBI identifiers, InChi identifiers, and labels are used to find matches in ChEBI." However, it is not clear to me what this means.

4. The biochemical terminology has been hard to understand at various places formr as a computer scientist who knows at least the ChEBI ontology. I guess the same will hold for the general NeSy audience. For example, I has to look up even a term occurring in the first sentence of the abstract, namely "digenic deletion fitness". Also, it was not clear to me that digenic deletion is an instance of gene interaction. Another example: the curves in Fig. 5 plot optical density (and in the caption, the curves are termed "groth curves"), but it is not explicitly mentioned in the paper that optical density is an indicator for measuring yeast growth.

detailed comments

p.2
"KGs like ... combines"

p.4, section 2.2
negative examples are pushed towards disjointness of boxes. Is this really intended?

p.7
You refer to Figure 4(a), probably you mean 3(a).
Figure 3: it would help to show the diagonals.

p.8
The subfigures in Figure 4 should be named (a) and (b).

**Anonymity:**

Remain anonymous